# Development and Characterization of PBSA-Based Green Composites in 3D-Printing by Fused Deposition Modelling

**DOI:** 10.3390/ma15217570

**Published:** 2022-10-28

**Authors:** Babacar Niang, Nicola Schiavone, Haroutioun Askanian, Vincent Verney, Diène Ndiaye, Abdoulaye Bouya Diop

**Affiliations:** 1Laboratory of Atmospheric and Ocean-Material Sciences, Energy, Device, Training and Research Unit of Applied Sciences and Technologies, Gaston Berger University, Saint-Louis 234, Senegal; 2National Centre for Scientific Research, Clermont Ferrand Institute of Chemistry, Clermont Auvergne University, SIGMA Clermont, 63000 Clermont-Ferrand, France

**Keywords:** natural fibers, biomaterials, 3D printing, fusion deposition modeling, rheology

## Abstract

Fused deposition modelling is a rapidly growing additive manufacturing technology due to its ability to build functional parts with complex geometries. The mechanical properties of a built part depend on several process parameters. The effect of wood content on the properties of 3D printed parts has been studied. Four types of filaments using poly(butylene succinate-co-adipate) (PBSA) with different reinforcement levels of Typha stem powder 0%, 5%, 10%, and 15% by weight were used for 3D printing. The density of the filaments and parts printed in this study increased with the Typha stem powder content. The thermal stability, mechanical performance, and viscoelastic properties of the different biocomposite filaments and 3D printed objects were analysed. The results show an increase in the crystallisation kinetics and a slight decrease in the thermal stability of the biomaterials. Compared to virgin PBSA FDM filaments, the PBSA biocomposite filament filled with Typha stem powder showed an increase in the tensile strength of the parts and specimens from 2.5 MPa to 8 MPa and in the modulus of elasticity from 160 MPa to 375 MPa, respectively, with additions of 5%, 10%, and 15% by mass. The addition of Typha stem fibres generated an increase in the elastic behaviour and relaxation time of the biomaterial structure, visualised by increases in the values of the viscosity components. The surface morphology reveals a decrease in the porosity of the printed samples.

## 1. Introduction

These last years, we can see a rising interest on the study of composite materials with natural fiber reinforcement, also known under the name of wood-plastic composites (WPCs) [1]. Moreover, the scientific community adopted the term “bio-composite” in order to classify the materials that are at least made of by one bio-based component, e.g., bioplastics, natural fibers, and nanocellulose. Biocomposites help to manufacture products with the required properties for a scale of applications [2], in accordance with the environmental norms [3].

These last years, we notice a high interest in fiber reinforced bioplastics; among the most used biopolymers are polylactic acid (PLA), poly(butylene succinate adipate) (PBSA), poly(hydroxybutyrate)(PHB), and polycaprolactone (PCL), which, because of their tremendous ability to reduce carbon footmarks, these biopolymers are considered as adequate resins to reduce global warming [4]. At the same time, scientists have strived to find the smartest alternatives for materials from renewable sources with synergistic properties. Consequently, natural fibers became the suitable option for academic and industrial research for distinct areas such as packaging, automotive, textiles, and building in reason of their advantages such as low cost, light weight, high strength/weight ratio, low density, and biodegradability [5]. In today’s world, natural fibers are rather well coherent to present needs, meeting a wide range of applications, from engineering material reinforcements to papermaking. For instance, there are significant university studies on cellulose’s structure and properties [6], together with lignocellulosic fibers, natural or chemically treated. The interest on these subjects is rising exponentially in the industrial applications [7].

These biocomposites are most commonly used in manufacturing processes such as extrusion, injection moulding, and thermoforming, but due to the new challenges of the industry, namely the manufacture of complex geometries and rapid prototyping, it is necessary to develop high-performance printable biocomposites.

A solvent blending approach was developed to ensure the uniform distribution of the binary biocomposites formed. Hot filament extrusions were performed by mixing hemicellulose and PLA with a variable ratio of up to 25% hemicellulose, the authors show a good print quality of the 3D scaffold prototypes [8].

The scientific work in this field imposes a need for continuous flow of the material, which leads to a disturbance during the printing process, to which is added a lack of anisotropy that significantly affects the properties of the analysed materials. Solutions are being sought to improve this method such as printing speed, layer deposition height, nozzle and platform temperature, filling pattern screen angle, construction orientation, and the addition of particles to the manufactured materials, particularly those based on polylactic acid [9,10]. Physico-chemical modification methods are used to improve the interfacial adhesion between the hydrophilic wood flour and the hydrophobic PLA matrix, including coupling agents and pre-treatment of the phases [11,12].

It is in this context that 3D printing brings several advantages in the manufacture of composites, including a high-precision, cost-effective, and customized geometry.

Among the many additive manufacturing techniques, molten deposition modelling (FDM—also known as Fused Filament Fabrication (FFF)) can be performed on thermoplastic polymers or composites, to allow a filament to be printed in layers. The main advantages of FDM are its low cost, relatively high speed, and potential to reinvent the design process [13]. Thermoplastic polymers reinforced with natural fibres (wood, flax, hemp, etc.) has a promising scale of specific mechanical properties [14] in combination with a reduced environmental imprint [15].

In this article we have chosen the PBSA because of it being a biodegradable plastic suitable to replace conventional plastics, that is, capable of satisfying the requirements of a daily use plastic in terms of functional and environmental properties. It naturally deteriorates in just a few years. PBSA is also known for its flexibility and can be obtained by polymerizing glycols with dicarboxylic acids.

We were interested in the Typha australis fibres (the Typhaceae family), which constitute one of the abundant plants rich in northern Senegal and, which, until today, remains little exploited in the field of biocomposites manufacturing.

The aim of our work is to develop new polybutylene succinate co-adipate biocomposites in combination with Typha stem fibres, which can be easily transformed by extrusion, are printed by the melted deposition method, and have good properties to consider multiple applications.

## 2. Materials and Methods

Polybutylene succinate co-adipate (PBSA) is synthesized by reaction of polycondensation of glycols and aliphatic dicarboxylic acids. Semicrystalline poly(butylene succinate-co-butylene adipate) (PBSA), referenced as PBE001, was supplied by NaturePlast (Ifs Clermont, France).

PBSA is a copolyester with a melting temperature of 90 to 120 °C. It can be formed by extrusion, like synthetic thermoplastic polymers. Its glass transition temperature is in the range of −45 to −10 °C and its density is 1.25 g/cm^3^. The high chemical and thermal resistance and biodegradability of PBSA are properties that promote its use in a wide range of applications. The natural charge used was the Typha stem powder. It was picked on the Senegal River banks, in Bango in the region of Saint Louis. The plant was cut and dried in the sun then underwent a mechanical grinding to transform the stems into powders. Le diamétre de la tailles des particules varie entre 21.8 μm et 598 μm.

### 2.1. Filament Extrusion and 3D Printing

Filaments for 3D printing were manufactured based on the techniques described by Filgueira et al. [16]. A Noztek filament extruder was used. The applied temperatures in the Noztek extruder were 140, 150, and 140 °C. The filaments were used for 3D printing in an Original Prusa i3. Standard dog bone specimens for mechanical testing were 3D printed (Figure 1).

Extrusion and Printing Conditions in Table 1 and Table 2. 

### 2.2. Mechanical Characterization

Mechanical characterization of the studied materials was realized with tensile stress tests. Thanks to the tests, we were able to evaluate the mechanical performance of the selected composite material in comparison with the virgin polymer and in relation with the manufacturing process used to create the test pieces. Tensile test was undertaken under the ASTM D638 standard at a 48–54% of relative humidity with a speed of 1.2 mm/min and a temperature of 23 °C with 48 h of preparation.

### 2.3. Thermogravimetric Analysis TGA

Thermogravimetric analysis (TGA) of all the patterns was fulfilled by Q500 (TA Instruments) under the nitrogen atmosphere at the flow rate of 60 mL/min. Samples of mass ranging from 7 to 13 mg were placed on the platinum pan and were heated from room temperature to 600 °C at the heating rate of 10 °C/min. Thermogravimetric analysis (TGA) of all the patterns was fulfilled by Q500 (TA Instruments, New Castle, DE, USA) under the nitrogen atmosphere at the flow rate of 60 mL/min.

### 2.4. Differential Scanning Calorimetry (DSC)

We used differential scanning calorimetry (DSC) to carry out a Q 2000 (TA instruments), operated under a nitrogen atmosphere at the flow rate of 50 mL/min. The samples, weighing from 4 to 6 mg, were first heated to 200 °C from room temperature at a heating rate of 10 °C/min, and then were cooled from 200 °C to 40 °C at a cooling rate of 10 °C/min

### 2.5. Porosity Content Measurement

The porosity content was determined according to ASTMD638. The weight and volume of the composites were measured and the percentage of porosity was determined using the following Equation (1):(1)$$vp=1-\rho_{c}\left(\frac{1-w_{f}}{\rho_{m}}+\frac{w_{f}}{\rho_{f}}\right)$$
where *vp* is the porosity content; *ρ_c_*, *ρ_m_*, *ρ_f_*, is the composite, matrix, and fibre densities, respectively; and *w_f_* is the fibre weight ratio. Results were double-checked using image analysis of composite cross sections.

## 3. Results and Discussion

We realized the development process of the composite material with a PBSA matrix and a Typha flour fibre by examining different filling percentages (10, 20, and 30%), to determine the measurable configuration for the filament extrusion process.

### 3.1. Tensile Mechanical Properties

The tensile response of the materials studied can be seen in Figure 2 and Figure 3. A difference in the response of the virgin polymers compared to the compounds clearly appears. We notice that the tensile young modulus increases considerably with the reinforcement rate from 180 MPa to 350 MPa compared to the virgin matrix, this increase is due to the reinforcing character of the fibres, which induces a very important gain in the rigidity of the material.

However, the elongation at break is significantly higher for pure PBSA than for biocomposites, which is to be expected, because PBSA has a ductile matrix and the strain at break is a matrix-dominated property in this case.

Tensile strength of virgin polymer is higher than that of printed samples. At the other side, with the increase of natural fibres to the polymer, an increase in maximum strength from 3 MPa to 7.5 MPa was observed, respectively, for composites containing 5, 10, and 15 % reinforcement (Figure 3). This is attributed to the improvement of the interfacial characteristics between the constituents. During extrusion, an interface between the fibre and the matrix is formed. This interface is reinforced by reprocessing the molten compound during the printing process, the printing parameters have a significant influence on the mechanical properties of the parts and the printing orientation, maximising the filaments at 0°, i.e., aligned with the load direction in tensile tests, and the layer thickness of the printed filaments had a strong impact on the tensile strength of the printed parts [17,18]. Among the parameters, the nozzle temperature (Tn) is the one that has the most significant effect on the quality and strength of the printed specimens, as the viscosity of the extruded material depends directly on it [19]. In fact, we have set this temperature at 195 °C when printing the samples, which results in a drop in viscosity, which then induces transverse flows of the material after deposition, thus diminishing its roughness. This not only improves the surface quality, but also the bond strength between layers, as there is a better diffusion across the interfaces.

#### Porosity and Morphological Characterisation

Figure 4 below shows the fracture surfaces of the virgin polymer samples and the composites containing 5 to 15% reinforcement. We can observe on the polymer the absence of surface voids and there could be the presence of pores in volume, as shown by the calculation of the percentage of porosity. However, with regard to the fracture surfaces the reduction in micropores leads to an increase in the mechanical properties in traction, as the zones of initiation and the propagation of cracks are generally found in the zone subjected to deformation in traction and are closely linked to the presence of porosities (Table 3).

In addition, the decrease in porosity is a result of the flow characteristics of the extruded materials and is governed by the melt temperature, print speed, and certainly the composition and aspect ratio of the Typha stem powder.

The optimal printing speed and melting temperature, as well as the aspect ratio of the reinforcements can minimize voids in the parts [16].

In our biomaterials, the effect of porosity within the biocomposites depends on the type and direction of loading. The influence of porosity is greater on the shear and transverse strengths, while the properties influenced by the reinforcement factor are affected to a lesser extent. This is due to the fact that porosity reduces the effective cross-section of the matrix [20]. The printing direction and the weft angle are important for a good orientation and dispersion of the fibre in the polymer matrix.

### 3.2. Thermal Properties of a WCPBSA Filament

Before FDM printing, the thermal characterization of a filament is essential because the analysis of its thermal behaviour is able to provide information on the properties of a printed component. Figure 5 shows the thermograms of the Wood PBSA filament and wood powder samples as a function of temperature. These tests consist of measuring the change in the mass percentage of composites as a function of an upward gradient of temperature. We observe that the thermo-degradation of composites is done in three steps: the first step concerns the loss of mass less than 2% between the temperatures of 100 °C and 200 °C, corresponding mainly to the evaporation of water and non-combustible chemicals on the surface of wood particles.

The second stage involves the chemical decomposition of celluloses, hemicelluloses, and lignin that begin at low speeds at about 180 °C and peak at around 300 °C. They reported during the second stage, the total decomposition of hemicellulose and cellulose and the partial decomposition of the lignin. The final step is polymer degradation, which begins at approximately 320 °C and peaks between 400 °C and 500 °C) [21,22].

These results showed that the thermal degradation of wood fibres is minimal during the FDM printing process with an extrusion temperature between 195 and 200 °C, which is supposed to produce a WCPBSA component.

#### DSC Measurement

In the FDM printing process, the WPC filament was melted in a 210 °C heater and cooled on a 35 °C construction platform. Simulating this situation and understanding the properties of the FDM on the printed part, the effect of wood fibre on the thermal behaviour of the PBSA matrix in the WPC filament were determined by DSC measurements. Generally, the properties of polymeric products are influenced by the formation of crystalline structures and crystalline indices during heating and cooling processes. Figure 6 shows the heat flow of the fused PBSA filament (pure PBSA) and WPC filament at the second heat and cooling scan. It can be seen that the glass transition temperature of the PBSA was approximately −40 °C for all samples during the second heating process. When wood fibres were added to the PBSA (Figure 6), the cold crystallization exotherm was observed in the range of 15 °C to 50 °C at the second heating. Consequently, three peak melting temperatures were obtained at 86, 87, and 90 °C from the DSC curve of the WPC filament (Table 4).

It is clear from Figure 6 that the addition of Typha fibres to the PBSA results in an increase in the crystallization temperature T c of the samples. This can be explained by the nucleation capacity of the Typha fiber for the crystallization of PBSA. As the amount of added fibres increases, T c and the crystallization enthalpy (ΔH c) of the PBSA phase increase with the addition of Typha fibres, indicating that the fibres accelerate the crystallization process. The most important effect of wood on the semi-crystalline structure polybutylene succinate co-adipate (PBSA), is its ability to act as a nucleation agent that promotes crystallization.

As shown by the two upper curves (second heating) in Figure 6, there is a slight difference of about one percent between melting temperatures. The existence of this difference between the two melting temperatures has already been observed. This difference is probably due to the reorganization in the melted state, to the variation of crystalline dimensions or the presence of crystallizers of different stabilities [23,24].

### 3.3. Rheological Characterization

In order to further investigate the effect of the addition of Typha fibres on the rheological and structural behaviour of the materials, Cole–Cole diagrams were used (Figure 7). In this diagram, the imaginary viscosity component (η″) is plotted against the real viscosity component (η′). The graph should look like a semi-circle if the system describes a single relaxation. In heterogeneous melts containing agglomerated fibres of Typha stem, the semicircle shape of the Cole–Cole graph is modified, and the elastic component of the viscosity and the relaxation time increases [25].

The Cole–Cole plots of pure PBSA revealed a semicircle related to the single relaxation time. On the other hand, the addition of Typha stem fibres produced a growth in the elastic behaviour and relaxation time of the biomaterial structure, visualised by increases in the values of the viscosity components. This behaviour indicates the presence of agglomerated Typha stem fibres and decreases with the layer-by-layer printing method. In the light of these results, we can say that the continuous extrusion and 3D printing processes produce a better dispersion of Typha stem fibres within the polymer matrix. The presence of fillers affects the rheological behaviour of the polymer system. In the process of determining the rheological properties of biomaterials, it is important to take into account the type, size, and shape of the filler as well as its size distribution and concentration [26]. Moreover, it is important to highlight that the addition of Typha stem fibre did not alter the characteristic pseudoplastic behaviour of pure PBSA [27], which is a determining element in the development of biomaterials for the 3D impregnation process. Due to the very low glass transition of the virgin matrix, Typha stem powder leads to an increase in viscosity, which is very important for the possibility of manufacturing parts with complex geometry. In this Cole–Cole diagram, the biocomposites show shear thinning behaviour, i.e., the viscosity is a decreasing function of the shear rate. Figure 8 shows the samples (A1 and B1) one by one and in sets of three at the same time (A2 and B2, B3). In Figure 8 A1, the sample does not show a clear printed cubic geometry. This is due to the fact that the glass transition of the virgin PBSA is below −40 °C and also the melting temperature of the PBSA is well above 140 °C while the printing temperature is 195 °C. On the other hand, for the samples printed in sets of three, virgin PBSA has a much-improved cubic geometry. This could be explained by the longer printing time that allows the crystals to cool down before a new layer is applied. The same observation was made for biocomposites filled with Typha stem fibres. In addition to this, it can be clearly seen that Typha stem powder improves the viscosity of the biocomposites, which is shown by the much clearer cubic geometries in Figure 8 A2, B2, B3. Typha stem improves the mechanical and viscoelastic properties of the biomaterials and stabilises the geometry of the printed samples at high temperature levels.

### 3.4. Charpy Impact Tests

The results of the Izod impact test are presented in Figure 9. A decrease in the impact resistance of pure PBSA is observed with increasing the content of Typha stem fibres up to 5% Typha stem powder in the biomaterials. Indeed, this slight decrease is already reported for lignocellulosic fibre-reinforced polymer composites [28]. The mechanisms of impact energy absorption by the fibres are threefold, i.e., unbinding, tearing, and fracture, but due to the direction of printing and the screen angle of the fused deposition method, there is a good dispersion and distribution of the fibres in the polymer matrix and a decrease in absorption energy. This adhesion between the matrix and the fibre predisposes the material to a lower energy absorption. This leads to an improvement in the mechanical properties of the biomaterial. At 10% Typha stem fibre, the impact energy of the fibres increases, which shows that an optimal fibre concentration and an increase in impact properties have not yet been achieved [29,30]. This additional increase is due to a complete encapsulation of the fibres by the matrix. The reinforcement of the Typha stem powder in the composite leads to a decrease in the stress concentration due to the good adhesion between the Typha stem flour and the polymer. Crack propagation becomes difficult in biomaterials (Figure 10).

## 4. Conclusions

PBSA filaments reinforced with Typha stem fibres were successfully produced and then printed using a melt deposition modelling 3D printer. The results showed an improvement of the crystal structure by increasing the nucleation phenomenon, the rheological analysis shows a predominance of the viscous behaviour over the elastic behaviour of the printed biomaterials. The complex viscosity shows shear thinning behaviour and pseudo-plasticity of the mixtures. Interesting mechanical properties are observed in 3D FDM printed samples. For example, the strength properties and stiffness increase with the filler content in the biocomposites. The printability problem of virgin PBSA at high temperatures is improved by an increase in filament roughness and complex viscosity. In addition, there is a reduction in intra-filament porosity observed in the morphological analysis by the fracture surfaces. Overall, the results suggest that Typha stem could be successfully used to produce biocomposite filaments for FDM applications. These materials offer the opportunity to rapidly produce 3D degradable biocomposite prototypes for future applications in the consumer, automotive, and construction sectors.

## Figures and Tables

**Figure 1 materials-15-07570-f001:**
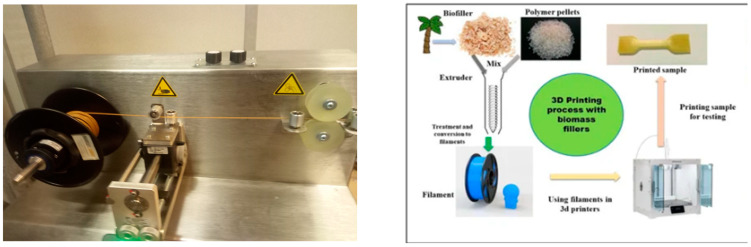
Schematic representation of a typical fused deposition modelling (FDM) device.

**Figure 2 materials-15-07570-f002:**
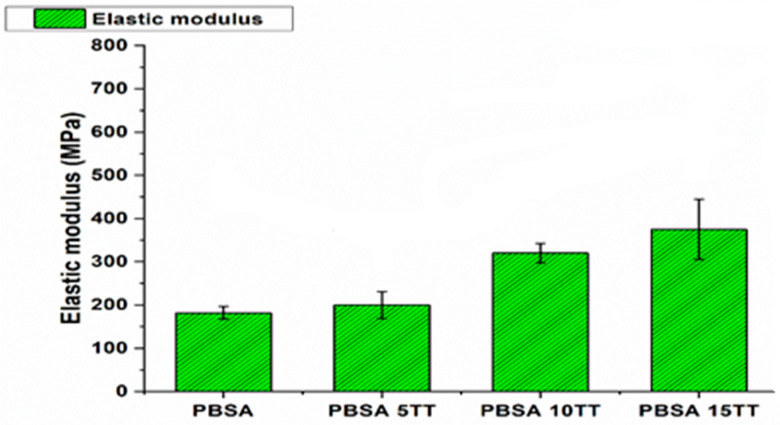
The tensile modulus of pure PBSA and its bio composites.

**Figure 3 materials-15-07570-f003:**
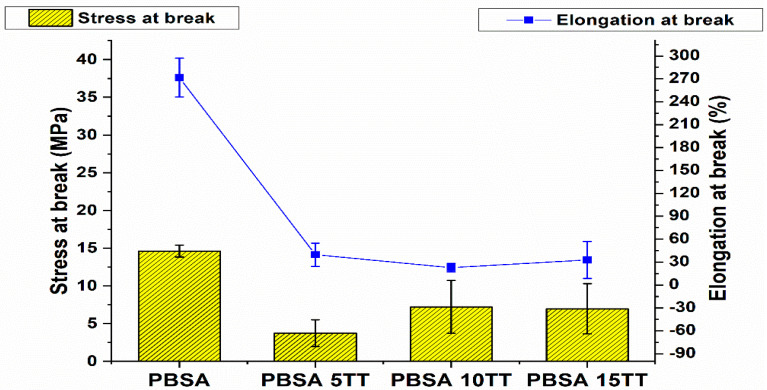
Tensile strength and maximum strain at break of pure virgin PBSA and biocomposites.

**Figure 4 materials-15-07570-f004:**
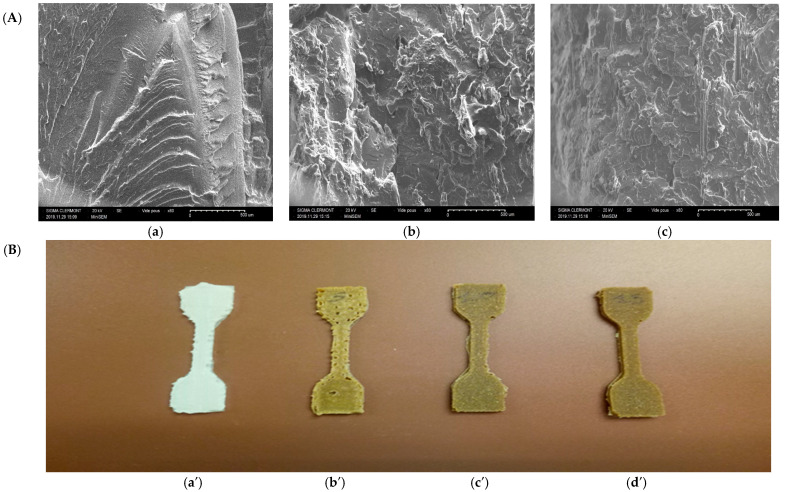
(**A**): SEM micrographs of biocomposites. (**a**) PBSA/5 TT, (**b**) PBSA/10 TT, and (**c**) PBSA 15 TT. (**B**): Printed specimens of virgin PBSA and biocomposites. (**a****′**) PBSA, (**b****′**) PBSA/5 TT, (**c****′**) PBSA/10 TT, and (**d****′**) PBSA/15 TT.

**Figure 5 materials-15-07570-f005:**
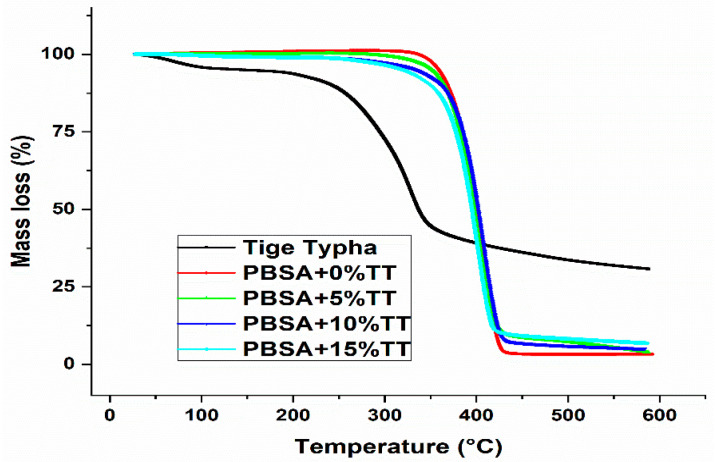
Thermogravimetry (TGA) of virgin PBS and its biocomposites.

**Figure 6 materials-15-07570-f006:**
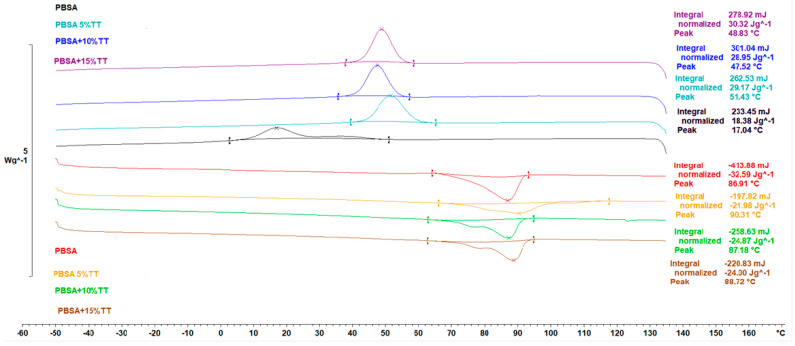
DSC thermograms of PBSA-based biomaterials.

**Figure 7 materials-15-07570-f007:**
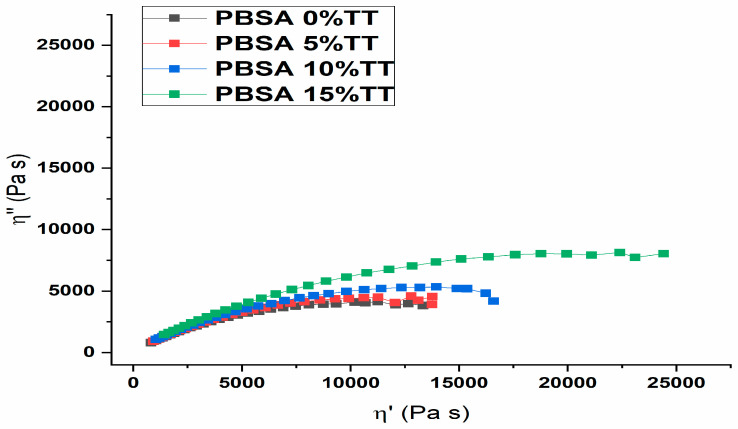
Complex plane diagrams for all the composites at T = 180 °C.

**Figure 8 materials-15-07570-f008:**
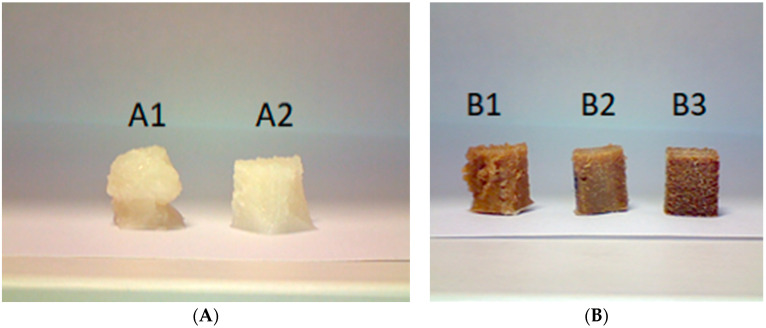
Cubic geometry of Typha-PBSA biocomposite.

**Figure 9 materials-15-07570-f009:**
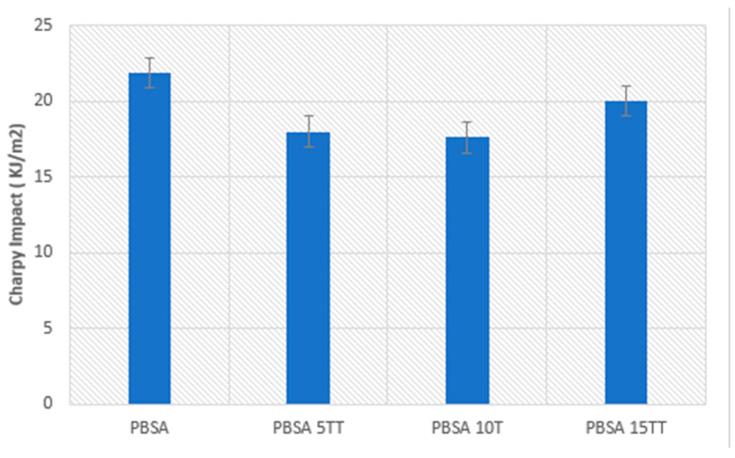
Charpy impact strength as a function of wood content.

**Figure 10 materials-15-07570-f010:**
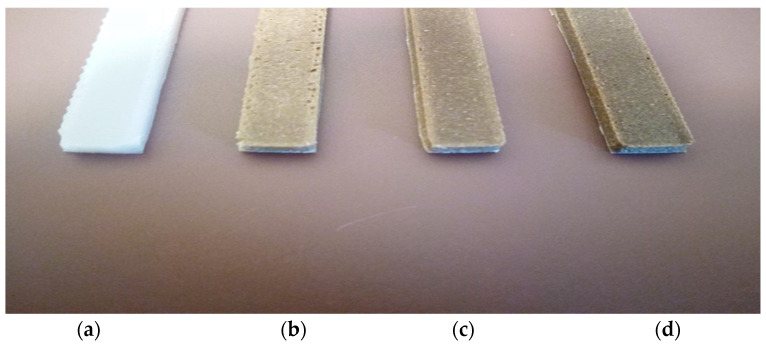
Charpy samples as a function of Typha fibre content. (**a**) PBSA, (**b**) PBSA/5 TT, (**c**) PBSA/10 TT, and (**d**) PBSA/15 TT.

**Table 1 materials-15-07570-t001:** Temperature of the different zones of the extruder and speed of rotation of the screw and collection of the filament.

T1	T2	T3	T4	T5	T6	T7	T8
140 °C	140 °C	140 °C	150 °C	150 °C	140 °C	140 °C	140 °C
The speed of rotation of the screw				200 rpm			
Filament collection speed				4.6 m/min			

**Table 2 materials-15-07570-t002:** Printing parameters.

Temperature	195 °C
Printing speed	30 mm/min
Printing speed	35 °C
Cooling fan	100%
Weft angle	45°

**Table 3 materials-15-07570-t003:** Degree of porosity as a function of reinforcement ratio.

Samples	0%TT	5%TT	10%TT	15%TT
porosity	26.5%	22%	18%	16%

**Table 4 materials-15-07570-t004:** Thermal properties of pure PBSA and its composites.

Samples	T_c_ (°C)	T_m_ (°C)	∆H_m_(J/g)	∆H_c_(J/g)
PBSA Pur	86.91 °C	17.04 °C	18.38	32.59
PBSA 5 TT	90.51 °C	51.43 °C	29.17	21.98
PBSA 10 TT	87.18 °C	47.52 °C	28.85	24.87
PBSA 15 TT	88.72 °C	48.83 °C	30.32	24.30

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
