# Peer review of "Development and Characterization of PBSA-Based Green Composites in 3D-Printing by Fused Deposition Modelling"

_materials, 2022, doi:10.3390/ma15217570_

Round 1
Reviewer 1 Report
The submitted article entitled "Development and characterization of PBSA -based green composites in 3D-printing by fused deposition modelling" presents the results of the research work focused on the development of new materials for additive manufacturing.
The subject of work seem to be novel, however some work has to be done before publication. The comment are presented below:
1. The introduction section should be more extensive. Since actually the polymer materials devoted for FDM technique are still based mostly on PLA it will be good to list mot popular modification techniques for PLA (polymer blends, nanoadditives, annealing).
The text contains many errors, in particular combined words, errors in labeling. Please make sure to awoid this kind of errors.
Quality (or resolution) of many figures is very low, prease increase the resolution for clarity
What was the matrix polymer (PBSA) type, was it purchased or self-made resin.
2.5 Porosity..... what was the ASTM standard method used during the measurements
fig.4 - the SEM pictures should be separated, actually it looks like the picture of one sample. Additionally, the SEM pictures should be described
fig.6 - DSC plots should be presented in more clear way, please group the plot separately for heating and cooling, eraze the sample weight and collect the numerical values in the table (peak temperature, enthalpy values, etc.)
Manuscript should be supplemented with the apperance of the dogbone specimens, the cubic samples suggest some major problems with the printability.
Author Response
Please find attached our response to the article

Reviewer 2 Report
The article reports on an interesting topic which recently attract significant attention from industry and research groups.
In general, the language needs attention, but the most important is the punctuation, spaces and formatting.
FFF does not refer to “molten filament manufacturing” but it denotes Fused Filament Fabrication.
The introduction is missing an adequate literature review that would allow the authors to identify the gap they want to bridge. Instead, the authors mentioned that “We are interested in in the Typha australis fibres (the Typhaceae family),…..” Please clearly review the most relevant work, highlight the scientific gaps and then link your work to address this scientific issue.
In the Materials and methods section, it is stated that “Le diamétre de la tailles des particules varie entre 21.8 μm et 598.μm.” What does it mean?
How the powder particles characterise?
The size of 598 µm is really large to be filled in the filament that can be printed afterwards, please comment.
Please rationalise the printing conditions applied.
Please add equation No. to equation in Section 2.5.
Some French words, phrases appear in different part of the paper, please proofread the paper.
The discussion of the results are good and sufficient but the quality of the Figures must be improved.
Conclusion needs improvements to highlight generic insight gained out of this research study.
Author Response
Please find attached our response to the article.
Reviewer 2

Reviewer 3 Report
The paper presents extensive characterization of PBSA -based green composites but some major revision of the English language and style has to be done a part from correcting the typos mistakes. For exemple :
- Introductrion --> delete the r
- many spaces between words are missing in the introduction
- "scientist shave"--> delete the « s »
- “3.1.1 porosity et morphology” change the French word “et” by and
So a global check of the English language is absolutely needed.
Best Regards
Author Response
Please find attached our response to the article
Reviewers 3

Reviewer 4 Report
1.Please revise carefully throughout, including the title, according to the format requirements of the journal Materials.
2. The abstract should be rewritten to highlight the main points of the study and interest the readers.
3.The resolution of Figure 1, 8 and 9 is too low.
4.Figures in Fig. 4 should be numbered as necessary and distinguished from each other.
5.Figure 6 should be redrawn and not produced directly using the test instrument.
6.The language of the full paper should be revised by an English-speaking industry insider.
Author Response
Please find attached our response to the article
Reviewers 4

Round 2
Reviewer 1 Report
There are still some unaddressed issues:
Porosity..... what was the ASTM standard method used during the measurements....while the D638 standard refers to tensile tests.
fig.4 - the SEM pictures should be separated, actually it looks like the picture of one sample. Additionally, the SEM pictures should be described
DSC plots should be presented in more clear way, please group the plot separately for heating and cooling
Author Response
You will find the answers in the attachments

Reviewer 2 Report
The Authors addressed the reviewer comments to a large extent but the responses are not clear. So, the authors are requested to provide proper responses to my comments.
For example
1) in the Materials and methods section, it is stated that “Le diamétre de la tailles des particules varie entre 21.8 μm et 598.μm.” What does it mean?
The answer is
In the particle size analysis of typha,
we found that Typha stem powder is characterised by particles of different sizes and geometry.
Indeed, the percentage of particles larger than 598 μm is approximately low.
6)Please rationalise the printing conditions applied.
the weft angle has been defined the printing conditions
The responses are not clear.
Author Response
Please find attached the responses to the comments

Round 3
Reviewer 1 Report
Most of my comments have been included in the text.
Reviewer 2 Report
The authors addressed the reviewers comments and the paper can be published after moderate English improvements.